# ProteinHypothesis: A Physics-Aware Chain of Multi-Agent RAG LLM for Hypothesis Generation in Protein Science

**Adib Bazgir & Yuwen Zhang** *
Department of Mechanical and Aerospace Engineering
University of Missouri-Columbia
Columbia, MO 65211, USA
{abwbw,zhangyu}@missouri.edu

**Rama chandra Praneeth Madugula**
Department of Mechanical Engineering
New York University
New York, NY 10012
{rm6057}@nyu.edu

## Abstract

Scientific hypothesis generation is fundamental to advancing molecular biology and protein science. This study presents a novel AI-driven multi-agent framework that integrates Retrieval-Augmented Generation (RAG) with structured experimental data for automated hypothesis generation and validation. The methodology employs scientific literature retrieval, structured dataset analysis, and multi-agent evaluation, ensuring that generated hypotheses are scientifically rigorous and experimentally testable. The framework consists of three key phases: (1) Hypothesis Generation, where insights from literature and structured data are synthesized using large language models; (2) Multi-Agent Evaluation through Chain of Thoughts (CoT) mechanism, where hypotheses are assessed for internal consistency, feasibility analysis, novelty assessment, scientific impact, and scalability/generalizability; and (3) Final Selection and Validation, where high-scoring hypotheses undergo refinement using protein-specialized agents and are linked to experimental validation strategies such as molecular dynamics simulations, site-directed mutagenesis, and structural characterization. Results demonstrate the system's ability to generate novel, high-impact hypotheses in protein stability, enzyme catalysis, ligand interactions, and biomolecular interactions, with broad applications in drug discovery, synthetic biology, and protein engineering. The study highlights the potential of AI-driven hypothesis generation in accelerating scientific discovery by integrating machine learning, structured data analysis, and multi-agent validation into research workflows. Our code is available at https://github.com/adibgpt/ProteinHypothesis.

## 1 Introduction

Hypothesis generation serves as the foundation of scientific inquiry, guiding researchers in formulating testable predictions that drive innovation and the expansion of knowledge across disciplines. Historically, this process has relied on human intuition, manual literature reviews, and structured methodologies based on logical reasoning and domain expertise. While these conventional approaches have led to groundbreaking discoveries, they are increasingly constrained by the limitations of human cognition, the exponential growth of scientific literature, and the rising complexity of interdisciplinary research. Scientists today face significant challenges in synthesizing vast amounts of information, identifying meaningful research gaps, and formulating hypotheses that integrate insights from multiple fields (Chai et al., 2024) (Abdel-Rehim et al., 2024). The limitations of traditional hypothesis generation methods are particularly evident in data-intensive disciplines such as biomedicine, astronomy, and computational sciences, where vast datasets and complex relationships make manual exploration impractical (Tong et al., 2024) (Ishikawa, 2024). As the scientific landscape grows more intricate, there is a critical need for automated and scalable methods to facilitate hypothesis generation. The advent of artificial intelligence (AI), and more specifically, Large Language Models (LLMs), has introduced a transformative approach to scientific reasoning.

---

*Corresponding Author: Yuwen Zhang at zhangyu@missouri.edu.

LLMs possess the ability to process extensive corpora of structured and unstructured data, extract latent patterns, and propose novel hypotheses with a breadth and depth beyond human capacity. These models, trained on diverse datasets, allow for the systematic generation of testable ideas, thereby reducing cognitive biases, mitigating information overload, and enhancing the efficiency of hypothesis-driven research (Takagi et al., 2023) (Pelletier et al., 2024). The emergence of AI-driven methodologies has redefined the research workflow by augmenting human intuition with computational power, enabling a more comprehensive exploration of scientific possibilities (Xiong et al., 2024) (Zhou et al., 2024). The integration of LLMs into scientific discovery has marked a paradigm shift from manual knowledge synthesis to AI-assisted, data-driven hypothesis generation. Unlike traditional heuristic-based approaches, LLMs leverage advanced reasoning techniques, retrieval-augmented knowledge synthesis, and structured workflows to generate hypotheses across diverse domains. Models such as GPT-4, BioGPT, SciBERT, and PMC-LLaMA have been instrumental in formulating novel research questions in fields ranging from biomedicine to materials science (Takagi et al., 2023). These models analyze vast knowledge repositories, incorporate real-time literature retrieval, and dynamically adapt to evolving research trends, allowing for continuous hypothesis refinement and optimization (Hu et al., 2024). One of the key innovations that have enhanced LLM-based hypothesis generation is Retrieval-Augmented Generation (RAG), which allows models to retrieve relevant scientific literature before generating hypotheses. This approach ensures that AI-generated hypotheses are contextually grounded in existing research, thereby increasing their validity and scientific relevance (Sybrandt et al., 2020) (Skarlinski et al., 2024). Additionally, multi-agent collaboration frameworks have been developed to simulate human-like brainstorming sessions, where different AI agents assume specialized roles such as hypothesis ideation, critique, and validation. These multi-agent systems, exemplified by ResearchAgent and The AI Scientist, enhance the robustness of AI-generated hypotheses by incorporating diverse reasoning strategies and iterative feedback loops (Park et al., 2024). The effectiveness of LLM-driven hypothesis generation is further amplified by iterative refinement frameworks, which involve a continuous cycle of hypothesis formation, evaluation, and improvement. Systems like Nova, ResearchAgent, and The AI Scientist utilize dynamic feedback mechanisms to assess the plausibility and novelty of hypotheses, refining them through multiple iterations before presenting them as viable research questions (Jamil et al., 2023) (Hu et al., 2024) (Lu et al., 2024). These iterative methodologies play a crucial role in mitigating common challenges associated with LLM-generated outputs, such as logical inconsistencies and speculative reasoning, ensuring that hypotheses align with domain-specific knowledge and empirical evidence (Ciucă et al., 2023) (Li et al., 2024).

The impact of LLM-driven hypothesis generation extends across multiple scientific domains, revolutionizing research in biomedicine, materials science, artificial intelligence, and the social sciences. In biomedical research, LLMs have been leveraged to propose new drug-target interactions, suggest therapeutic mechanisms, and uncover potential biomarkers for diseases such as cancer and neurodegenerative disorders (Qi et al., 2023) (Tadiparthi et al., 2024) (Sybrandt et al., 2020). Domain-specific models such as BioBERT and PMC-LLaMA enhance biomedical hypothesis generation by integrating curated datasets from PubMed, enabling AI systems to generate hypotheses that align with cutting-edge research (Zhou et al., 2024). In the field of materials science, AI-powered tools such as MOOSE-Chem and Nova facilitate the discovery of novel chemical compounds, optimize material properties, and predict interactions within molecular systems (Wang et al., 2024) (Liu et al., 2024) (Yang et al., 2024). These models utilize high-throughput screening and machine learning techniques to systematically explore vast chemical spaces, accelerating the development of advanced materials for energy storage, semiconductors, and sustainable manufacturing (Hu et al., 2024) (Park et al., 2024). Beyond the hard sciences, hypothesis generation has also been transformed in the social sciences and linguistics. Systems such as SciHypo and ResearchAgent have been applied to behavioral research, policy analysis, and linguistic studies, enabling the formulation of hypotheses on human behavior, economic trends, and language evolution (Ishikawa, 2024) (Koneru et al., 2023) (Bersenev et al., 2024). Despite their remarkable potential, LLM-driven hypothesis generation systems face several challenges. One of the primary concerns is hallucination, where LLMs generate speculative or unverifiable hypotheses that lack empirical grounding (Xiong et al., 2024) (Pelletier et al., 2024) (Jamil et al., 2023). Another significant limitation is bias in training data, which can lead to skewed insights (Proebsting & Poliak, 2024). Computational efficiency and scalability remain significant hurdles (Qi et al., 2024) (Yang et al., 2024) (Bersenev et al., 2024). Finally, ethical considerations, transparency, and reproducibility must be addressed to ensure AI-generated hypotheses align with scientific integrity (Park et al., 2024).

The current work distinguishes itself from existing AI-driven hypothesis generation approaches by introducing a three-phase multi-agent evaluation system that refines and experimentally validates hypotheses, rather than relying on single-stage generation. Unlike previous methods focused on biomedical and general scientific domains, the utilized approach in this study uniquely integrates structured physics-based data, particularly in protein science. A key innovation is the protein-specialized multi-agent framework, which leverages domain-specific agents (e.g., BioAgent, StrucAgent, EvoAgent, DrugAgent) to ensure biochemical, structural, and evolutionary relevance. Unlike prior heuristic-based models, this system employs Chain-of-Thought (CoT) reasoning for systematic hypothesis refinement, assessing internal consistency, feasibility, novelty, scientific impact, and scalability at multiple levels. Furthermore, this investigation explicitly links AI-generated hypotheses to experimental validation, incorporating techniques such as molecular dynamics simulations, site-directed mutagenesis, cryo-EM, and biophysical assays, enabling real-world applicability. Finally, the cross-disciplinary adaptability of this approach extends beyond biomedical research to drug discovery, protein engineering, synthetic biology, and biomolecular interactions, advancing AI-driven scientific inquiry with greater reliability, testability, and domain specificity.

## 2 PROPOSED APPROACH

The proposed system for scientific hypothesis generation integrates Retrieval-Augmented Generation (RAG), structured data embedding, and a multi-agent Large Language Model (LLM) pipeline. Through a series of automated and semi-automated steps, the framework gathers relevant literature, processes experimental data, creates rich vector embeddings, and formulates initial hypotheses. These hypotheses are then refined in multiple stages by specialized AI agents, with the ultimate aim of producing scientifically grounded and experimentally testable conclusions.

### 2.1 RETRIEVAL OF SCIENTIFIC LITERATURE AND STRUCTURED EXPERIMENTAL DATA

In the first phase, the system collects domain-relevant knowledge from two primary sources involving online repositories (e.g., arXiv) and structured experimental datasets (e.g., CSV files). This unified pool of raw information serves as the foundation for RAG-based hypothesis generation. The overall schematic of RAG system is illuminated in Figure 1.

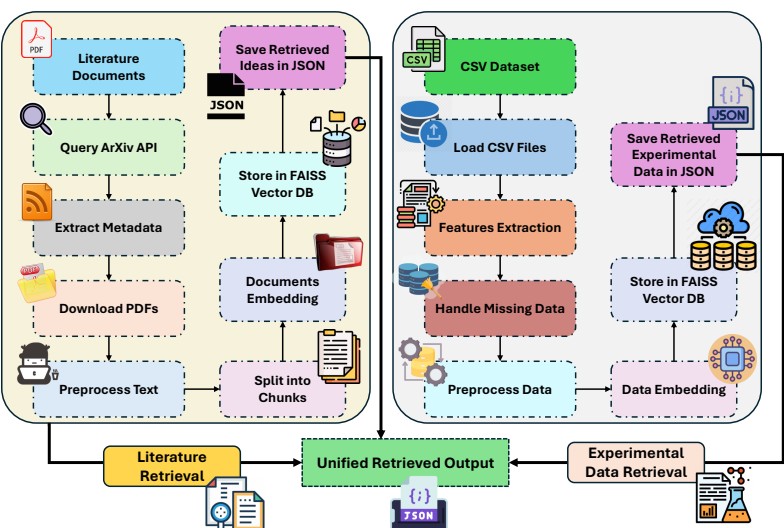

Figure 1: RAG system workflow for scientific document and experimental data retrieval.

### 2.1.1 LITERATURE RETRIEVAL AND PREPROCESSING

The pipeline queries large online repositories such as arXiv (Appendix A1.1) using domain-specific keywords (e.g., "Protein Science," "Protein Folding") and retrieves metadata—titles, abstracts, and

download links—through parsing tools. Where feasible, full-text PDFs are also retrieved, ensuring that all relevant content is available. Before embedding, raw text undergoes a cleaning and tokenization process to remove artifacts (e.g., HTML tags, excessive whitespace). At this stage, key ideas or segments can be stored in JSON for future reference, enabling quicker lookups or version control.

### 2.1.2 EXPERIMENTAL DATA RETRIEVAL AND PREPARATION

Experimental data, often in CSV format, is loaded via utility modules such as LangChain's CSVLoader. This data may describe protein sequences, protein properties, or other domain-specific protein science metrics (Appendix A1.2). Relevant features (e.g., structural attributes, numeric measurements) are then extracted, and missing data is handled either by imputation or exclusion, depending on scientific requirements. If desired, the processed dataset can be saved to JSON, facilitating cross-referencing and logging.

### 2.1.3 DOCUMENT AND DATA EMBEDDINGS

Once gathered, both textual documents and experimental datasets are transformed into high-dimensional embeddings. In practice:

- **Chunking and Splitting:** Long documents are divided into context-preserving segments using a tool like 'RecursiveCharacterTextSplitter', ensuring each segment remains self-contained and interpretable.
- **Vector Embeddings:** Textual chunks may be embedded with 'SentenceTransformers', while structured data (after feature extraction) can leverage embeddings such as 'GoogleGenerativeAIEmbeddings'.
- **FAISS Vector DB:** All resulting vectors are stored in 'FAISS' (Facebook AI Similarity Search), which supports fast similarity lookups across the combined literature-data space.

### 2.1.4 RETRIEVAL-AUGMENTED GENERATION (RAG)

With the unified repository in place, the system can respond to hypothesis-generation requests by querying FAISS for textual and structured data vectors relevant to the protein topic. The retrieved segments—rich in empirical and theoretical context—are passed to LLMs (GPT-4o and Gemini-1.5 Flash), which generate an initial set of hypotheses following a structured template that includes:

1. Background Insight from Literature
2. Pattern Identified from Structured Data
3. Novel Hypothesis Proposal
4. Experimental Validation Strategy

This ensures every hypothesis originates from robust, data-backed premises.

## 2.2 MULTI-AGENT EVALUATION AND HYPOTHESIS REFINEMENT

After the RAG module produces an initial set of hypotheses, the system enters multiple phases of evaluation and refinement. Each phase involves a distinct configuration of LLM agents that scrutinize or enhance the hypotheses, and the corresponding schematic is depicted in Figure 2.

### 2.2.1 INITIAL TO REFINED HYPOTHESES

This process includes two consecutive phases regarding the hypothesis generation as follows:

- **Phase 1:** Generates a preliminary hypothesis list from the retrieved literature and experimental data. These are stored in a minimal form, awaiting further validation.
- **Phase 2:** A "General Multi-Agent LLM" refines each hypothesis by clarifying assumptions, improving experimental designs, and suggesting complementary data references. The outcome is a more coherent and feasible hypothesis set. Five utilized general-purpose agents are namely Internal Consistency agent, Feasibility Analysis agent, Novelty Assessment agent, Scientific Impact agent, and Scalability/Generalizability agent.

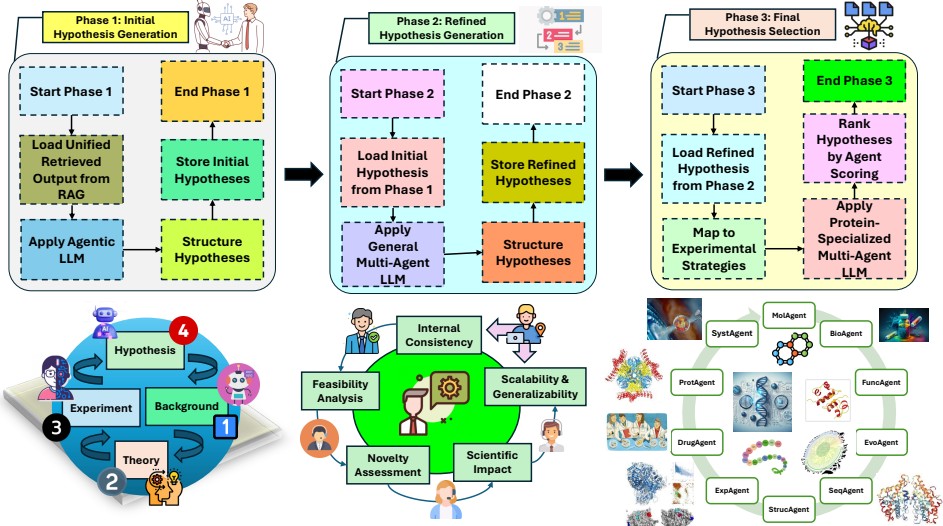

Figure 2: Chain of Multi-agent LLM systems workflow for three consecutive phases.

### 2.2.2 AGENT SCORING AND SPECIALIZED VALIDATION

This part of the hypothesis generation process pertains to rigorous refinement using specialized protein agents as discussed below:

- **Phase 3:** twelve Domain-focused protein agents (e.g., BioAgent, MolAgent, EvoAgent) evaluate each refined hypothesis according to criteria such as molecular stability, functional relevance, or potential for therapeutic applications. Each agent assigns a numerical score (often 1–3) on different aspects. Hypotheses failing to meet threshold scores in any major category are flagged for reprocessing, while those satisfying all criteria proceed to final selection.

### 2.2.3 CHAIN-OF-THOUGHT (CoT) REASONING.

Throughout the multi-agent review, Chain-of-Thought reasoning is used to trace each hypothesis's logical underpinnings. Agents or the system itself may detect contradictions, missing arguments, or alignments with known theories. This iterative, transparent reasoning process ensures that each hypothesis matures into a thoroughly vetted, testable proposition. Top-scoring hypotheses emerging from Phase 3 are designated as final and saved—along with evaluation logs, references, and code snippets—in JSON for reproducibility.

## 3 RESULTS AND DISCUSSION

The integration of RAG-Multi-Agent LLM system with structured physics-based datasets and scientific literature retrieval presents a novel approach for hypothesis generation. The results obtained from the scientific literature analysis and the structured experimental datasets provide complementary insights that collectively enhance the quality of generated hypotheses. In this section, we provide a detailed discussion of the extracted findings, emphasizing their relevance, reliability, and impact on hypothesis-driven research.

### 3.1 ANALYSIS OF RAG-BASED LITERATURE RETRIEVAL

The RAG-based literature retrieval methodology provides a structured approach for extracting scientific insights from peer-reviewed studies, enabling evidence-driven hypothesis generation. By systematically analyzing research, it identifies dominant trends, recurring themes, and inconsistencies in molecular biology and protein science, reinforcing established knowledge while uncovering

research gaps. A clear example of this methodology in action is illustrated in Figure 3, which showcases how the RAG system processes scientific literature retrieval. The Figure 3 highlights the structured query-response approach used to summarize key elements of a paper, including title, abstract, main hypothesis, and summary of results. This process ensures that the extracted insights are scientifically rigorous, thematically categorized, and directly relevant to hypothesis development. Key findings from RAG-based literature retrieval reveal several recurring themes:

- **Protein Stability and Folding** – Molecular interactions govern protein folding pathways and stability, making this a central topic in structural biology.
- **Computational Approaches** – Graph neural networks (GNNs) and transformer-based models play a crucial role in protein structure prediction, while computational studies provide insights into phase behaviors and crystallization dynamics.
- **Protein-Ligand Interactions** – Research highlights how small molecules modulate protein activity, emphasizing the role of conserved structural motifs in binding mechanisms.
- **Evolutionary Adaptations** – Studies demonstrate that protein sequences evolve while preserving core functions, underscoring the significance of conserved motifs in evolutionary biology.

Beyond trend identification, RAG-based retrieval effectively detects contradictions across studies. For instance, conflicting findings regarding mutation-induced protein stability emphasize the need for context-aware hypotheses that account for variables like pH, temperature, and cofactor availability. The example in Figure 3 further illustrates this by presenting how RAG systematically processes diverse research sources to uncover nuanced scientific debates. This methodology accelerates advancements in protein science, structural biology, drug discovery, and synthetic biology, reinforcing the transformative role of AI-driven literature analysis in modern research.

## 3.2 Analysis of Physics-Based Dataset Insights

The input file consolidates physics-based dataset insights, complementing literature-driven analysis by revealing key relationships between sequence, structure, and function in proteins. A clear example of this structured data retrieval process is illustrated in Figure 4, which demonstrates how the RAG system processes experimental data to extract meaningful insights. The Figure 4 highlights the step-by-step breakdown of column names, descriptive sentence construction, and summary extraction, showcasing how structured data can be systematically analyzed and contextualized to support hypothesis generation. A major finding is that secondary structure predictions align with specific sequence motifs (e.g., alpha-helices and beta-sheets), clarifying functional roles such as enzymatic catalysis and ligand binding. For instance, the protein `"3my2 A-P0ADV9"` exhibits predicted helices (H) and loops (L) at multiple labeled positions, reinforcing the link between folding patterns and biological function. The structured query-response methodology demonstrated in Figure 4 illustrates how computational tools facilitate precise functional annotation from sequence data. Several structural motifs correlate with functional traits:

- **Hydrophobic cores** contribute to protein stability.
- **Catalytic-site motifs** highlight key enzymatic mechanisms.
- **Conserved secondary structures** suggest evolutionary constraints, guiding mechanistic hypotheses.

By analyzing `"foldseek seq"` and `"label"` columns, researchers can identify shared features that aid in experimental validation, supporting mutagenesis studies and advanced simulations. The structured approach illustrated in Figure 4 ensures that such insights are systematically extracted and articulated, reinforcing computational predictions with biological relevance. A strong correlation exists between functional labels and structural features, such as metal ion-binding sites in flexible loop regions, which influence ion coordination. This allows for testable hypotheses, such as modifying loop residues to assess changes in binding affinity. Finally, the dataset highlights the variable functional importance of repeated motifs—some are highly conserved and essential, while others show variability with minimal functional impact. As demonstrated in Figure 4, structured data retrieval helps distinguish critical motifs from neutral variations, ensuring that hypothesis generation is precise, data-driven, and experimentally testable.

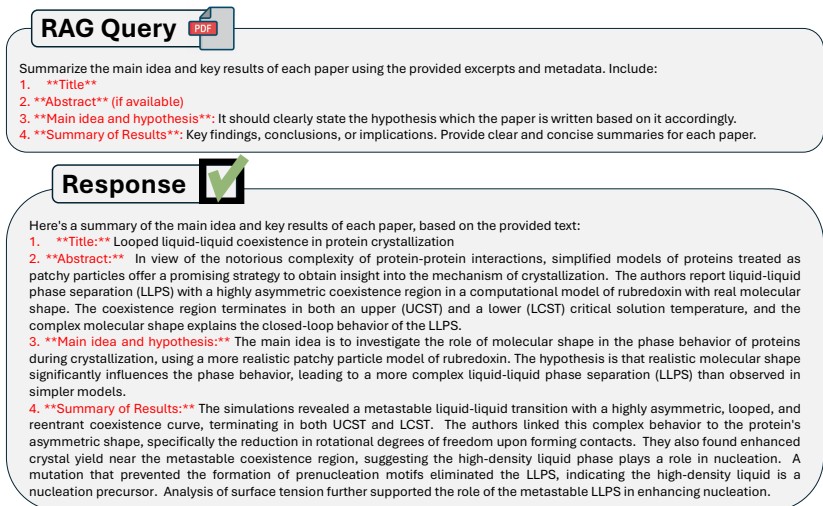

Figure 3: An example of RAG system output for scientific literature retrieval.

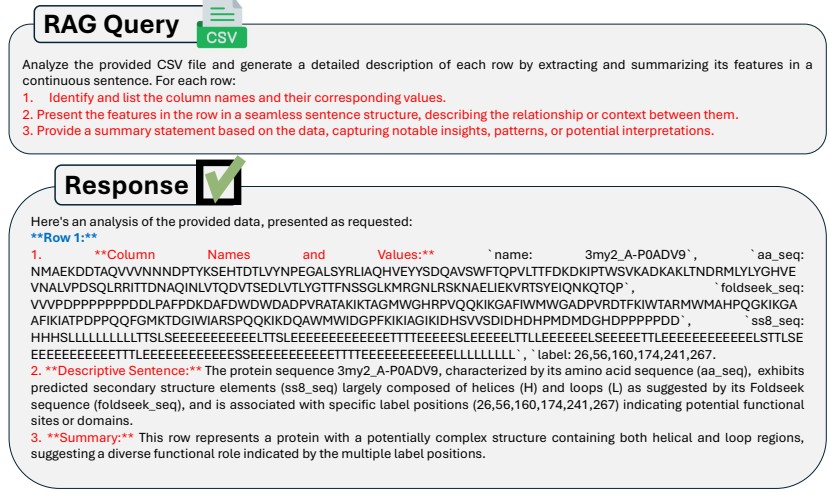

Figure 4: An example of RAG system output for experimental data retrieval in a single row.

### 3.3 1ST PHASE OF HYPOTHESES GENERATION USING MULTI-AGENT LLMS

The multi-agent system integrates literature-based insights with structured experimental data, ensuring a rigorous and systematic approach to hypothesis generation. This phase produces hypotheses based on empirical patterns, novel formulations, and validation strategies, ensuring scientific robustness. A detailed discussion of the prompt guiding this process is available in the Appendix (A2). Figures 5 and 10 illustrate the structured hypothesis generation process, where scientific literature and structured data are synthesized into novel, testable hypotheses. Examples include LLPS-related hypotheses integrating glycine-rich sequence data and protein flexibility hypotheses combining GNNs and hydrophobic patch analysis.

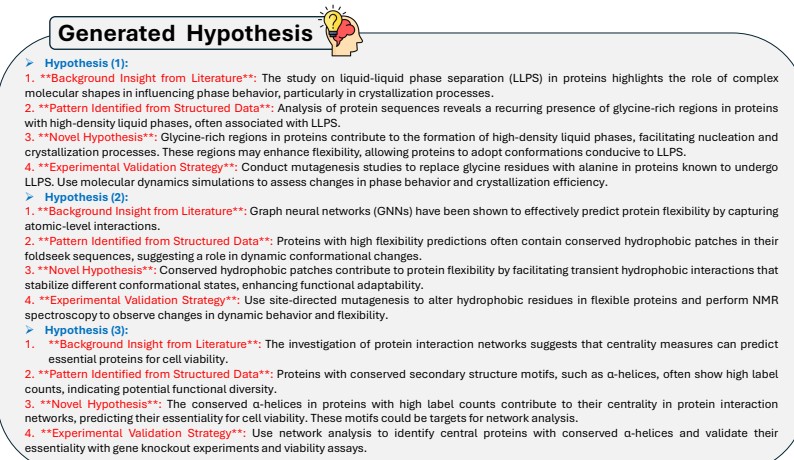

Figure 5: An example of generated hypotheses in the 1st phase of hypothesis generation.

## 3.4    2ND PHASE OF HYPOTHESIS GENERATION USING MULTI-AGENT LLMS

The second phase of the multi-agent system focuses on refining, validating, and ranking hypotheses generated in the first phase, ensuring scientific rigor through systematic Chain of Thoughts (CoT) assessments of internal consistency, feasibility, novelty, impact, and scalability. Each hypothesis undergoes logical coherence checks, experimental testability evaluations, and computational validation to enhance reliability. A detailed discussion of the prompt and the first and third levels of reasoning responses guiding this process is provided in the Appendix (A3).

Figure 6 and Figures 11-13 (Appendix (A3)) illustrate the structured refinement process, demonstrating how hypotheses are systematically evaluated, ranked, and selected based on scientific merit. These refined hypotheses establish strong correlations between secondary structure motifs and functional properties. Findings emphasize the role of $\alpha$-helices in stabilizing interactions, $\beta$-strands in cold-adapted proteins, hydrophobic patches in protein aggregation, and aromatic residues in structural stabilization. Additionally, $\beta$-hairpin motifs are linked to protein knot formation, reinforcing the evolutionary significance of these structural elements.

The refinement process also highlights novel insights across protein evolution, nanotechnology, and neurodegenerative diseases. As depicted in Figure 6 and Figures 11-13, studies explore disulfide bonds in evolutionary stability, amphipathic helices in membrane curvature, and hydrophobic regions in amyloid formation, offering implications for drug delivery, biomaterials research, and disease modeling. These findings pave the way for further experimental validation and translational applications in molecular biology and biophysics.

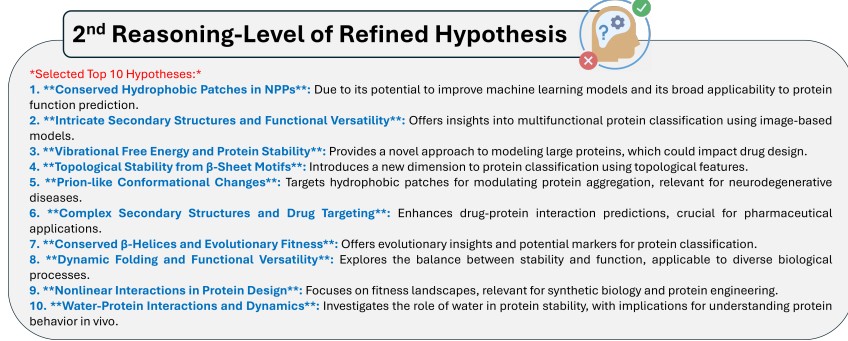

Figure 6: Top 10 generated hypotheses in the 2nd phase of hypothesis generation.

### 3.5 3RD PHASE OF MULTI-AGENT SYSTEM ANALYSIS: FINAL HYPOTHESIS SELECTION AND EXPERIMENTAL VALIDATION

A specialized multi-agent framework evaluates hypotheses from various scientific perspectives, ensuring rigorous assessments based on biochemical plausibility, drug discovery relevance, evolutionary consistency, functional applicability, and structural compatibility. Each agent plays a distinct role in refining and validating hypotheses. BioAgent ensures that hypotheses align with biochemical principles, enzyme kinetics, and protein stability. DrugAgent explores pharmaceutical relevance, assessing how hypotheses contribute to drug discovery and therapeutic applications. EvoAgent evaluates sequence conservation trends, verifying evolutionary consistency and structural adaptation. FuncAgent examines the functional impact of hypotheses on protein interactions, enzymatic functions, and cellular mechanisms. ProtAgent focuses on protein engineering applications, ensuring hypotheses contribute to protein design and synthetic biology. SystAgent integrates insights from systems biology, analyzing hypotheses within biological networks and large-scale molecular interactions. MoLAgent investigates molecular-level interactions, ensuring the structural and chemical feasibility of proposed hypotheses. ExpAgent assesses the experimental feasibility of hypotheses, mapping them to existing laboratory techniques and methodologies. SeqAgent evaluates sequence-function relationships, identifying conserved motifs and patterns essential for protein function. StrucAgent verifies structural compatibility, ensuring that hypotheses align with known protein folding, stability, and molecular architecture. As can be observed in Figure 7, top 2 hypotheses are selected by applying protein-specialized agents into 10 generated hypotheses from the second phase of multi-agent hypothesis generation process. According to Figure 7, this approach is able to generate both "More General" and "More Specific" hypotheses referring to 'Hypothesis 1' and 'Hypothesis 2', respectively. Accordingly, the 'Hypothesis 1' mainly focuses on general aspects of proteins and how they influence on protein interactions, stability, and function, while the 'Hypothesis 2' further explores the protein function and applicability by directly incorporating the physics-based experimental datasets (beta-helix motifs and a protein type in the "Class 0") into the generic form of hypothesis and make it more physically grounding for domain-knowledge experts. The corresponding prompt used for generating top 2 selected hypotheses is provided in Appendix (A4).

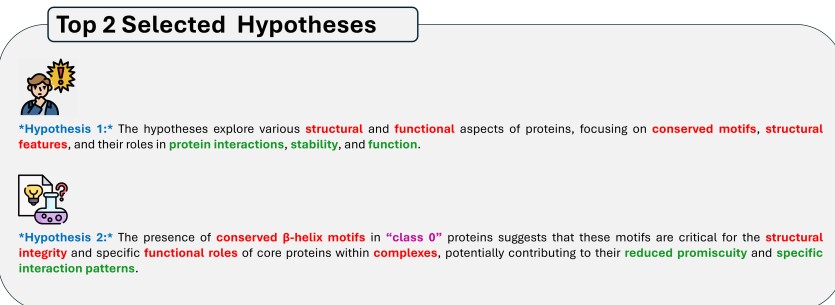

Figure 7: Top 2 selected hypotheses in the 3rd phase of hypothesis generation.

## 4 CONCLUSION

This study presents a novel AI-driven multi-agent framework for scientific hypothesis generation, integrating Retrieval-Augmented Generation (RAG), structured experimental data, and multi-agent validation to formulate, evaluate, and refine hypotheses in molecular biology and protein science. By combining scientific literature retrieval with structured physics-based datasets, the system enables the synthesis of data-driven, experimentally testable hypotheses, offering a scalable and automated approach to hypothesis-driven research. Results demonstrate the system's capability to generate novel, high-impact hypotheses related to protein stability, ligand interactions, enzyme catalysis, and biomolecular networks, with applications in drug discovery, synthetic biology, and protein engineering. The integration of multi-agent AI evaluation enhances the reliability of generated hypotheses, ensuring alignment with scientific principles, experimental feasibility, and broader biological relevance.

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

# A  APPENDIX

## A.1  UTILIZED DATASET

The dataset utilized in this study integrates retrieved scientific documents and structured experimental datasets to enable hypothesis generation through a physics-aware, multi-agent RAG framework. The retrieval process involves collecting relevant literature from online repositories and extracting structured data from domain-specific CSV files. This section details representative examples of both sources to illustrate how they contribute to AI-driven hypothesis formation.

### A.1.1  SAMPLE DOCUMENTS USED FOR RETRIEVAL

The retrieval process involves querying large-scale scientific repositories, such as arXiv, bioRxiv, and other publicly available research databases, using domain-specific keywords related to protein science, hypothesis generation, and computational modeling. A sample of the documents retrieved and processed is presented in Figure 8, showcasing the first page of representative papers used for document retrieval. These documents cover a broad spectrum of protein science topics, including:

- **Protein adsorption mechanisms** – Understanding how proteins interact with surfaces and their role in biomolecular interactions.
- **Protein-protein interactome refinement** – Using gene expression data to analyze functional relationships between proteins.
- **Protein hypernetworks** – A logic-based framework for understanding dependencies and perturbations in protein interaction networks.
- **Protein folding and molecular dynamics** – Investigating how sequence-structure-function relationships impact protein stability and flexibility.

Each retrieved document is processed through metadata extraction, content segmentation, and vector embedding to ensure that the RAG system can efficiently identify relevant insights for hypothesis generation.

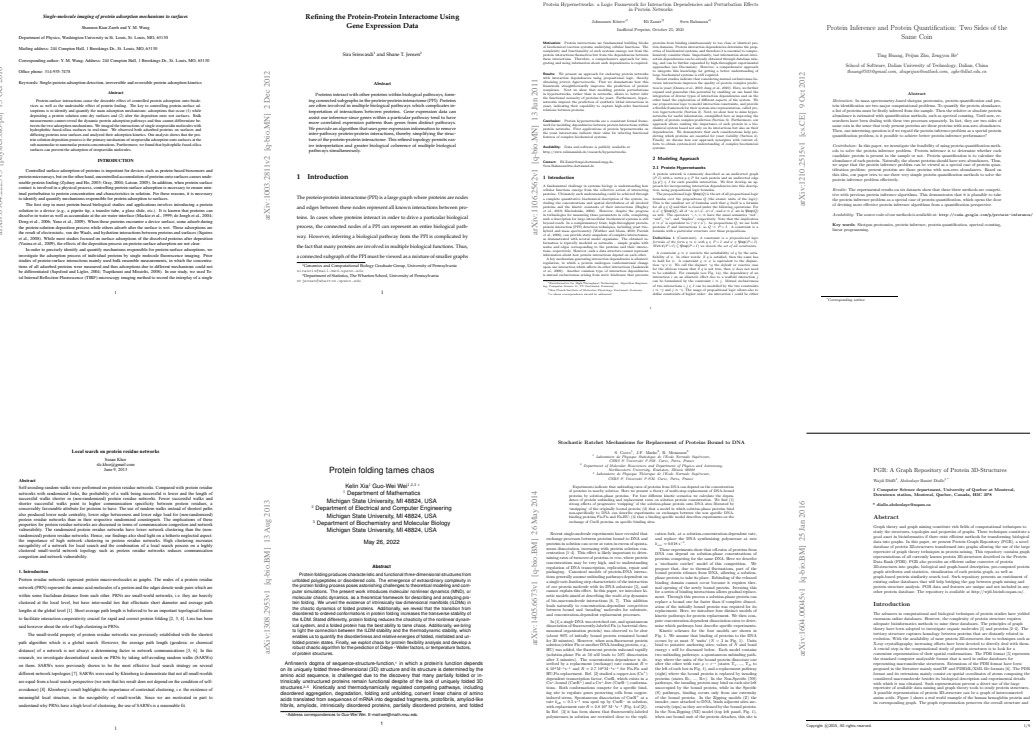

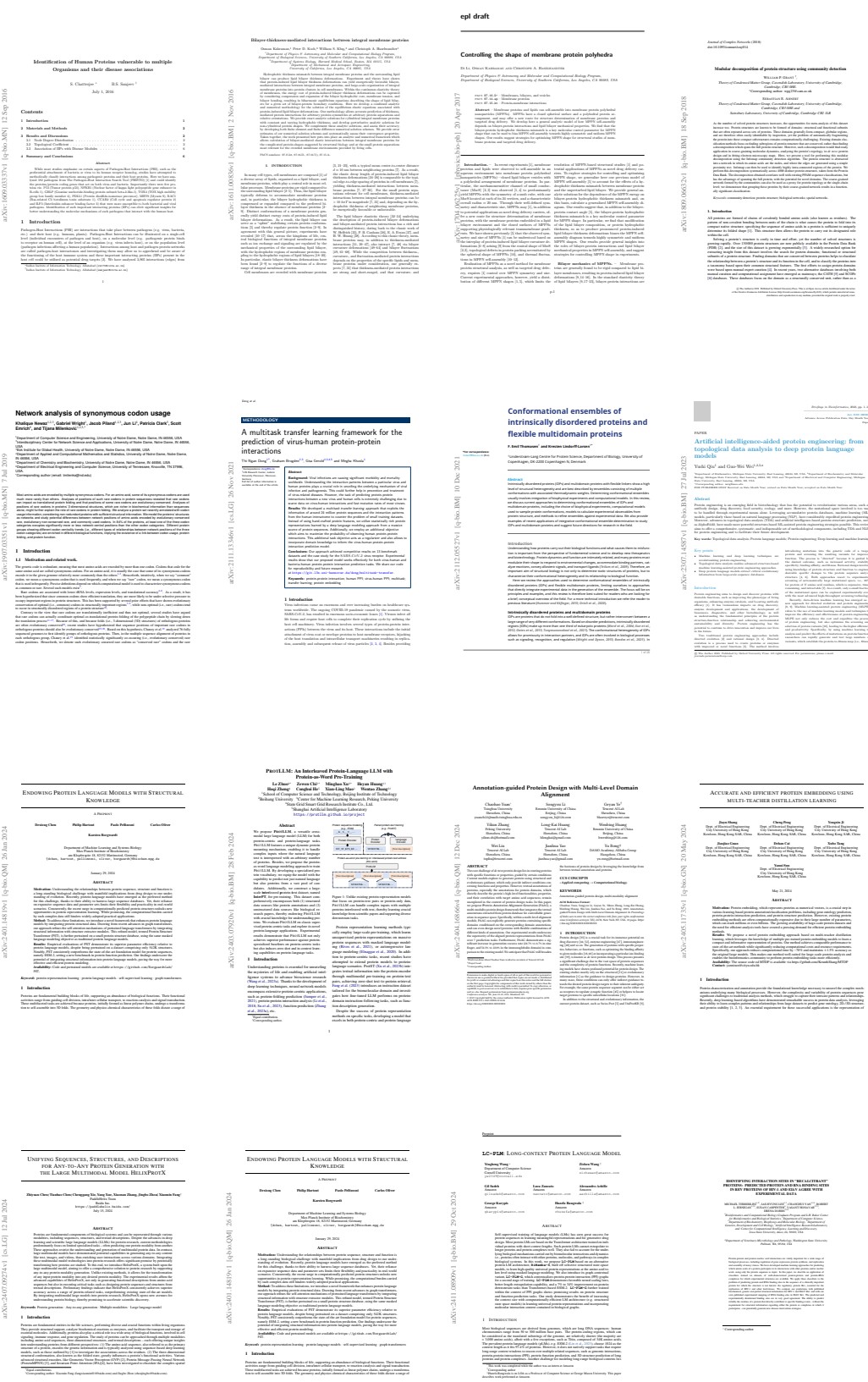

Figure 8: Sample documents used for protein science literature retrieval.

### A.1.2 SAMPLE CSV FILES USED FOR RETRIEVAL

In addition to literature retrieval, the framework integrates structured experimental datasets in CSV format to enhance the empirical grounding of generated hypotheses. Figure 9 presents example rows from structured datasets that include:

- **Protein sequence data** – Amino acid compositions and sequence motifs relevant to secondary structure prediction.

- **Structural properties** – Annotations of *alpha-helices, beta-strands, and loop regions* derived from experimental datasets.

- **Functional classifications** – Labels indicating biological significance, enzymatic activity, and evolutionary conservation.

- **Experimental measurements** – Physicochemical attributes such as stability metrics, binding affinities, and solubility factors.

These structured datasets are processed using feature extraction, data embedding, and similarity search techniques to enable dynamic hypothesis refinement. The integration of document-based insights with structured experimental data ensures that hypotheses are formulated with both theoretical and empirical rigor.

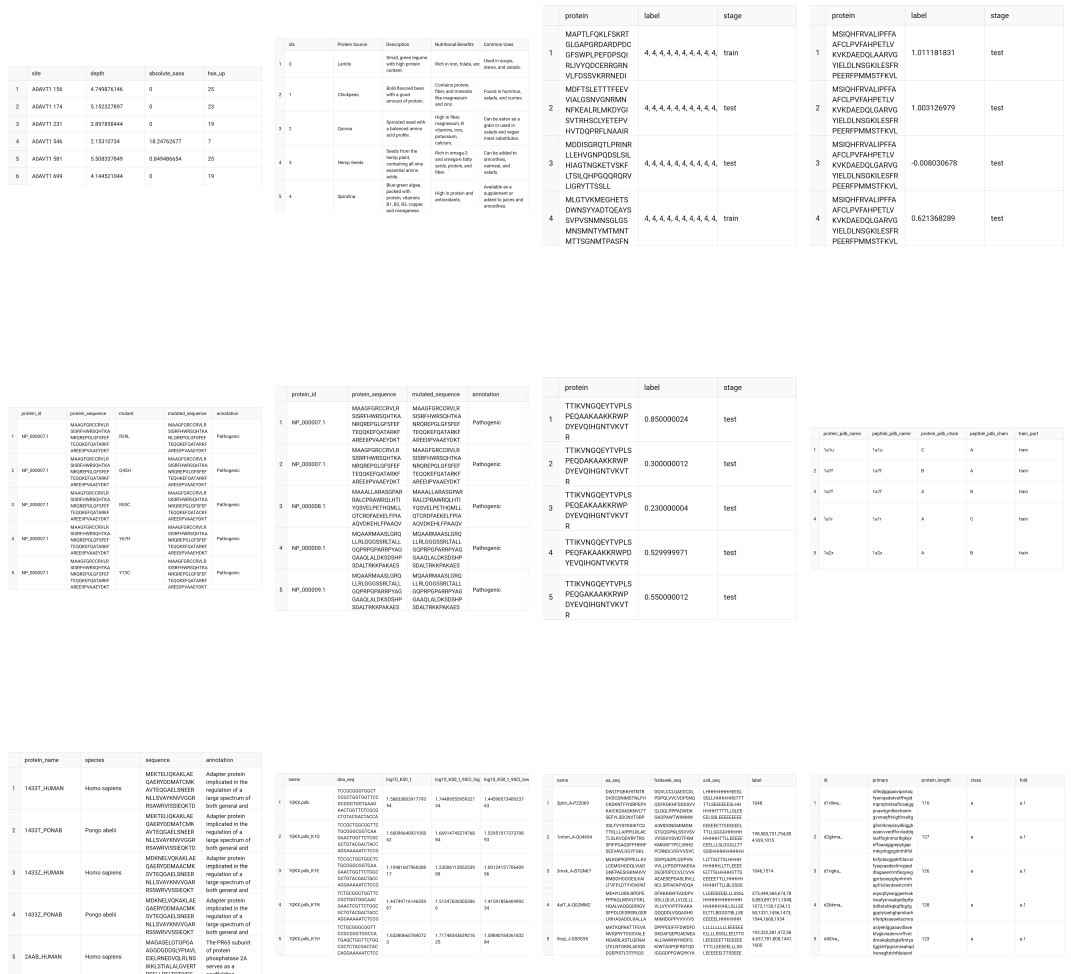

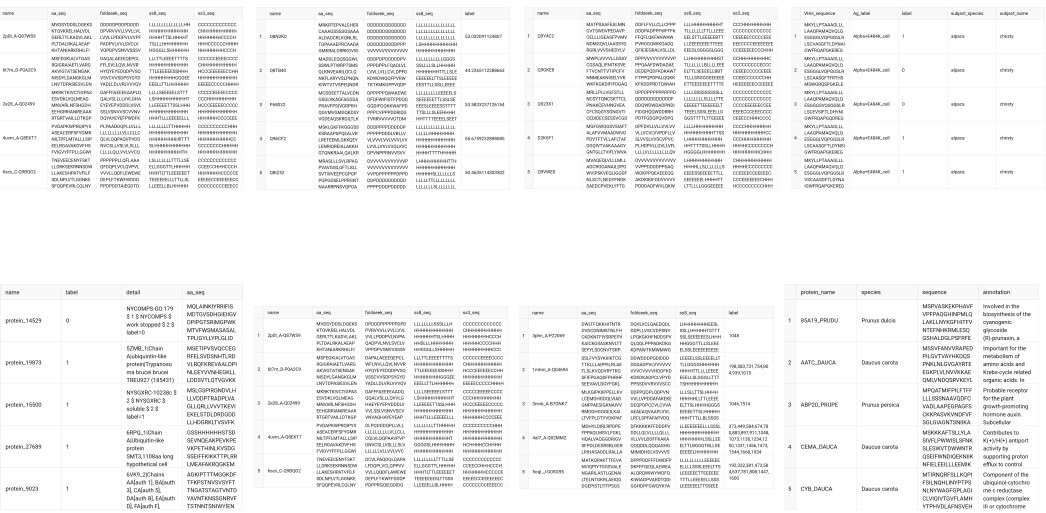

Figure 9: Sample CSV files used for experimental data retrieval.

## A.2   1ST PHASE OF HYPOTHESES GENERATION USING MULTI-AGENT LLMS

The hypothesis generation prompt in Figure 10 provides a structured framework for integrating scientific literature insights with structured experimental data to formulate novel protein science hypotheses. This prompt is designed for a multi-agent system, ensuring that generated hypotheses are scientifically grounded, data-driven, and experimentally testable.

The process of hypothesis generation in this phase follows a four-step structure:

1. **Background Insight from Literature** – The prompt instructs the system to extract key scientific principles, mechanisms, or trends from peer-reviewed research. This ensures that hypothesis formation is anchored in established knowledge.

2. **Pattern Identified from Structured Data** – The system analyzes structured experimental datasets to identify relevant sequence motifs, secondary structure correlations, and functional site patterns. This step ensures that hypotheses are empirically supported.

3. **Novel Hypothesis** – Based on the previous insights, the system proposes a new hypothesis that explicitly combines literature-derived knowledge with structured data analysis. This integration fosters innovation and scientific discovery.

4. **Experimental Validation Strategy** – To ensure that hypotheses are testable, the prompt mandates the inclusion of specific experimental techniques such as molecular dynamics simulations, mutagenesis, and crystallography. This enhances the practical applicability of each hypothesis.

Additionally, the prompt highlights the importance of structured data by requiring the analysis of amino acid sequence motifs, secondary structure correlations, and functional site patterns. It also provides examples of integrating structured data, such as identifying conserved hydrophobic patches and assessing the impact of mutations at labeled sites.

By enforcing this systematic approach, the prompt ensures that hypotheses are rigorously formulated, scientifically relevant, and experimentally testable, making it a powerful tool for AI-driven hypothesis generation in protein science.

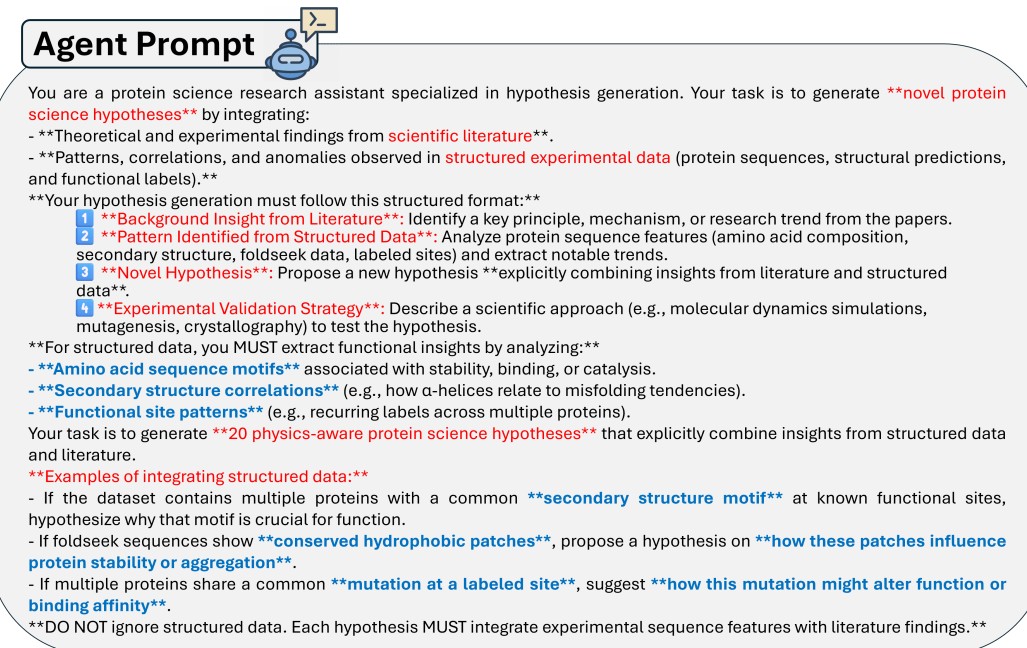

Figure 10: The utilized prompt for 1st Phase of Hypotheses Generation using Multi-Agent LLMs.

### A.3 2ND PHASE OF HYPOTHESIS GENERATION USING MULTI-AGENT LLMS

The hypothesis evaluation process in the second phase, as illustrated in Figure 11, follows a structured multi-step reasoning framework known as Chain of Thought (CoT). This approach ensures that each hypothesis undergoes systematic refinement based on predefined scientific criteria. The CoT General Multi-Agent Prompt is designed to assess hypotheses across five fundamental aspects:

1. **Internal Consistency Check** – This step verifies whether the hypothesis logically follows from established scientific principles and does not contradict existing biochemical and structural knowledge.

2. **Feasibility Analysis** – Hypotheses are assessed for their experimental testability by determining whether existing methodologies or computational models can validate them.

3. **Novelty Assessment** – The prompt evaluates whether the hypothesis introduces a unique or underexplored concept, ensuring its contribution to new scientific knowledge.

4. **Scientific Impact** – The broader implications of the hypothesis are examined, including its relevance to advancing fundamental research, biomedical applications, or translational science.

5. **Scalability and Generalizability** – This step determines whether the hypothesis extends to related proteins, biological systems, or molecular contexts, ensuring its applicability beyond a singular case.

The output format of the evaluation follows a structured approach to ensure clarity and reproducibility. As can be seen in Figure 12, each hypothesis is summarized in an Initial Hypothesis Summary, followed by a Step-by-Step Evaluation that applies the five CoT criteria. After this process, the system selects the Top 10 Hypotheses based on their scientific rigor and experimental feasibility. Finally, an Experimental Validation Strategy is provided, outlining computational or laboratory approaches for hypothesis testing.

By employing a multi-agent framework with Chain of Thought reasoning, the system systematically refines and ranks hypotheses, ensuring that only the most scientifically promising and experimentally viable hypotheses progress to further validation. This structured approach enhances the reliability,

reproducibility, and impact of AI-driven hypothesis generation in protein science and molecular biology.

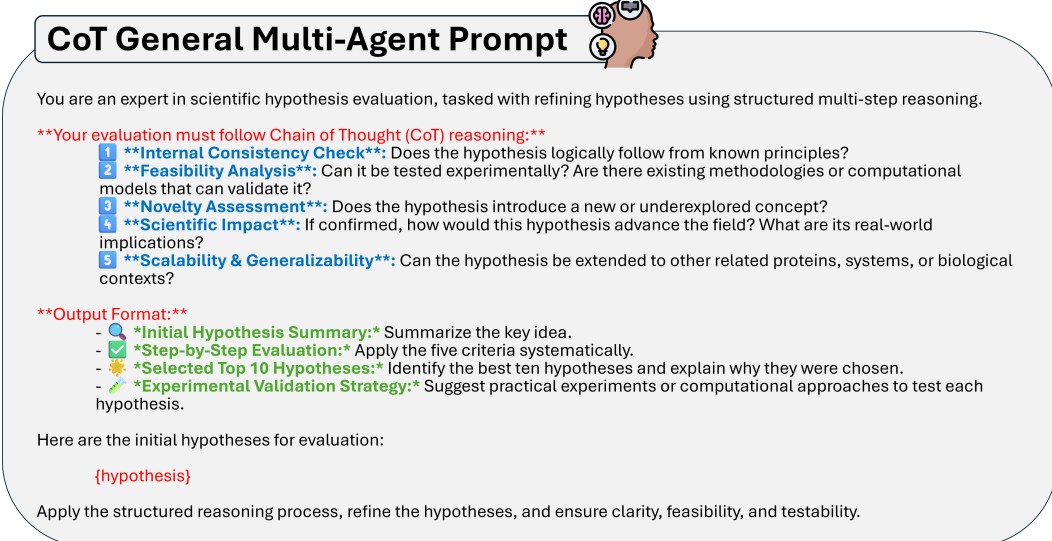

Figure 11: The prompt used for Chain of Thought (CoT) reasoning in the 2nd Phase of Hypothesis Generation using Multi-Agent LLMs.

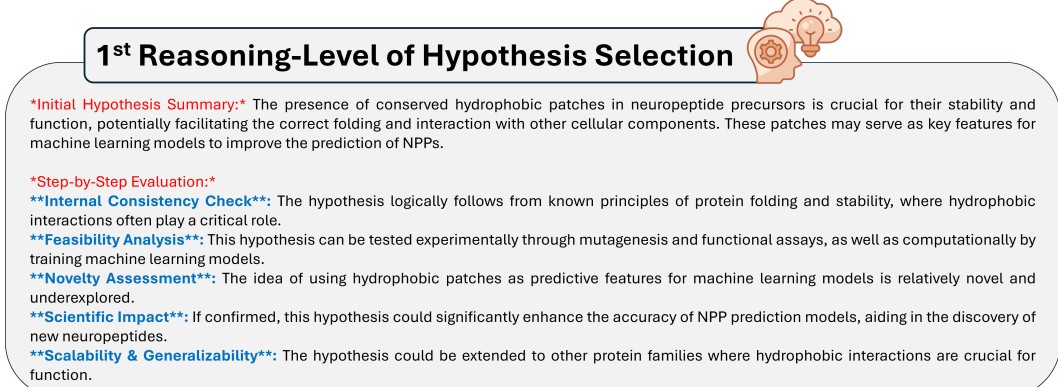

Figure 12: The 1st reasoning-level of hypothesis selection output in the 2nd Phase of Hypothesis Generation using Multi-Agent LLMs.

Figure 13 presents the third reasoning level of hypothesis evaluation in the 2nd phase of hypothesis generation using Multi-Agent LLMs. At this stage, hypotheses undergo rigorous experimental validation to ensure their feasibility and scientific credibility. The multi-agent system systematically assigns validation strategies to each hypothesis, aligning them with computational and laboratory-based experimental techniques.

The Experimental Validation Strategy involves a diverse set of methodologies tailored to assess different aspects of structural and functional protein properties. For instance, hypotheses related to hydrophobic patches in neuropeptides undergo mutagenesis studies to alter hydrophobic residues, with circular dichroism spectroscopy and functional assays providing empirical validation. Similarly, intricate secondary structures and functional versatility are evaluated through molecular dynamics simulations, deep learning-based image classification, and conformational flexibility analyses.

Additional experimental strategies include vibrational free energy and protein stability assessments, which utilize molecular dynamics simulations with vibrational free energy calculations, and validation through X-ray crystallography and NMR spectroscopy. Hypotheses concerning topological stability from beta-sheet motifs are examined using atomic force microscopy, applying persistent homology for structural analysis. In the context of prion-like conformational changes, in vitro aggregation assays coupled with fluorescence spectroscopy enable tracking of conformational alterations.

To validate hypotheses with applications in drug targeting and molecular interactions, Bayesian neural networks are trained on structural features and subsequently tested using binding assays. Similarly, hypotheses related to conserved alpha-helices and evolutionary fitness undergo phylogenetic analysis correlated with stability assays and machine learning models for classification.

Further validation strategies include the use of stopped-flow spectroscopy to investigate protein folding kinetics and its correlation with functional assays, as well as molecular dynamics modeling for nonlinear protein interactions, which are subsequently validated through thermal denaturation assays. Lastly, water-protein interactions and dynamics are explored through NMR spectroscopy and molecular dynamics simulations to understand their role in protein stability and function.

By integrating these diverse validation approaches, the multi-agent system ensures that each hypothesis is robustly tested using both experimental and computational methods. This final step in the evaluation process enhances the reliability of AI-driven hypothesis generation, paving the way for empirical validation and real-world scientific applications.

**3rd Reasoning-Level of Testifying Hypothesis**

*Experimental Validation Strategy:*
**1. Conserved Hydrophobic Patches in NPPs:** Conduct mutagenesis to alter hydrophobic residues and assess changes in stability and function using circular dichroism spectroscopy and functional assays. Train machine learning models with and without these features to evaluate predictive power.
**2. Intricate Secondary Structures and Functional Versatility:** Use molecular dynamics simulations to study conformational flexibility and implement image-based deep learning models for classification.
**3. Vibrational Free Energy and Protein Stability:** Perform molecular dynamics simulations with vibrational free energy calculations and compare with X-ray crystallography or NMR spectroscopy data.
**4. Topological Stability from β-Sheet Motifs:** Use atomic force microscopy to measure mechanical stability and apply persistent homology for topological analysis.
**5. Prion-like Conformational Changes:** Conduct in vitro aggregation assays and use fluorescence spectroscopy to monitor conformational changes.
**6. Complex Secondary Structures and Drug Targeting:** Train Bayesian neural networks on structural features and validate with binding assays.
**7. Conserved α-Helices and Evolutionary Fitness:** Perform phylogenetic analysis and correlate with stability assays, using machine learning for classification.
**8. Dynamic Folding and Functional Versatility:** Use stopped-flow spectroscopy for folding kinetics studies and correlate with functional assays.
**9. Nonlinear Interactions in Protein Design:** Model interactions using molecular dynamics and validate with thermal denaturation assays.
**10. Water-Protein Interactions and Dynamics:** Conduct molecular dynamics simulations and use NMR spectroscopy to study protein dynamics and stability.

Figure 13: The 3rd reasoning-level of hypothesis evaluation output in the 2nd Phase of Hypothesis Generation using Multi-Agent LLMs.

## A.4 3RD PHASE OF MULTI-AGENT SYSTEM ANALYSIS: FINAL HYPOTHESIS SELECTION AND EXPERIMENTAL VALIDATION

Figure 14 illustrates the deployed prompt for protein-specialized Multi-Agent systems in the third phase of hypothesis generation using Multi-Agent LLMs. This phase focuses on the final selection and experimental validation of the most promising hypotheses, ensuring their robustness across multiple scientific perspectives. The system employs a structured evaluation framework, where hypotheses are assessed by specialized agents, each responsible for a distinct scientific criterion.

The **Molecular Stability & Folding Agent** evaluates whether the hypothesis aligns with established principles of protein folding, stability, and aggregation. The **Biochemical Plausibility Agent** ensures that the hypothesis conforms to known biochemical principles, including enzyme kinetics

and ligand interactions. The **Functional Relevance Agent** assesses the biological applicability of hypotheses, determining their significance in cellular functions.

To validate hypotheses from an evolutionary standpoint, the **Evolutionary Consistency Agent** examines whether the proposed mechanisms align with evolutionary biology principles. The **Sequence-Function Relationship Agent** investigates how sequence variations influence protein function, identifying conserved motifs and their structural relevance. Similarly, the **Structural Compatibility Agent** ensures that the hypothesis is consistent with known 3D protein structures.

The **Experimental Validation Agent** determines whether the hypothesis can be empirically tested using available laboratory techniques, mapping it to established experimental methodologies. In the pharmaceutical domain, the **Drug Discovery & Therapeutic Potential Agent** evaluates the potential translational impact of the hypothesis for drug development and therapeutic applications. The **Protein Engineering & Synthetic Biology Agent** assesses the applicability of the hypothesis in protein design, synthetic biology, and biomolecular engineering.

Lastly, the **Interaction Network & Systems Biology Agent** evaluates the relevance of the hypothesis in the broader context of protein interaction networks and cellular systems. Each agent returns an evaluation score based on predefined scoring criteria, ensuring a systematic ranking of hypotheses. This structured approach enables an objective and multi-faceted assessment, facilitating the selection of hypotheses with the highest scientific merit and experimental feasibility.

By integrating these specialized assessments, the Multi-Agent system ensures that only the most robust and scientifically viable hypotheses proceed to experimental validation. This final phase enhances the reliability of AI-driven hypothesis generation, supporting advancements in structural biology, drug discovery, and protein engineering.

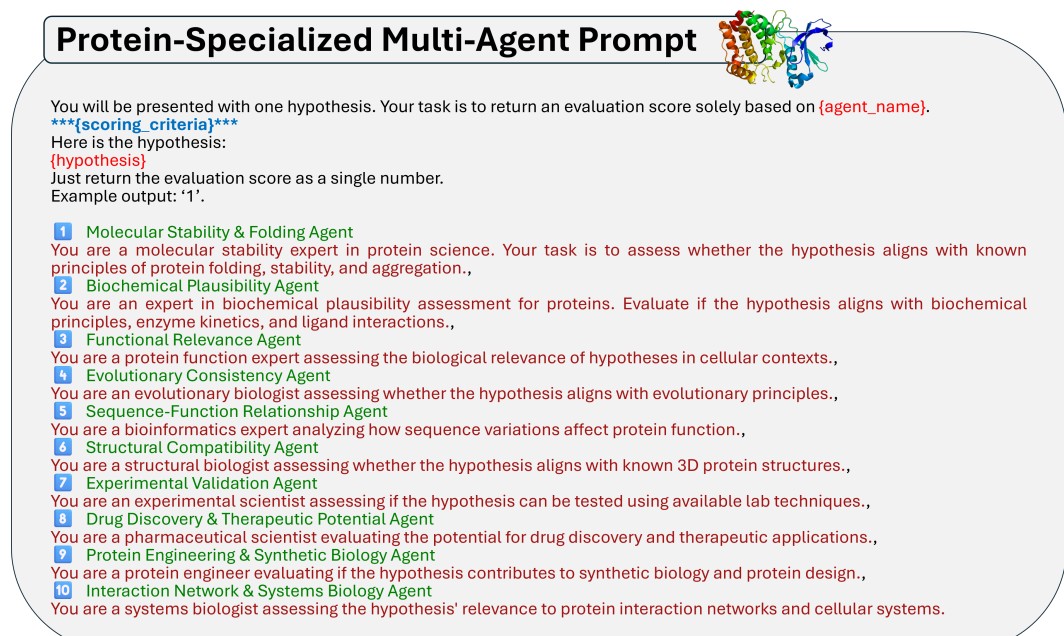

Figure 14: The deployed prompt for protein-specialized Multi-Agent systems in the 3rd Phase of Hypothesis Generation using Multi-Agent LLMs.

