# OpenReview forum: "ProteinHypothesis: A Physics-Aware Chain of Multi-Agent RAG LLM for Hypothesis Generation in Protein Science"
_ICLR.cc/2025/Workshop/AgenticAI — ICLR 2025 Workshop AgenticAI Poster_

### Official Review · Reviewer_obSo · 2025-03-02
**Review of ProteinHypothesis**

**Rating:** 5
**Confidence:** 5

**Review:**

# Summary:
This work introduces an AI-driven multi-agent framework for automated scientific hypothesis generation and validation in molecular biology and protein science. The framework integrates Retrieval-Augmented Generation (RAG) with structured experimental data and follows three key phases: (1) Hypothesis Generation, leveraging large language models to synthesize insights from literature and structured data; (2) Multi-Agent Evaluation, using a Chain of Thought (CoT) mechanism to assess hypotheses for consistency, feasibility, novelty, impact, and scalability; and (3) Final Selection and Validation, refining high-scoring hypotheses with protein-specialized agents and linking them to experimental validation strategies like molecular dynamics simulations and site-directed mutagenesis. The results demonstrate the system's ability to generate high-impact hypotheses in protein stability, enzyme catalysis, and biomolecular interactions, with applications in drug discovery, synthetic biology, and protein engineering. This study highlights AI’s potential in accelerating scientific discovery by integrating machine learning, structured data analysis, and multi-agent validation into research workflows.

# Strengths:
1. The paper presents a novel multi-agent framework integrating Retrieval-Augmented Generation (RAG) with structured experimental data, enhancing automated hypothesis generation and validation in molecular biology and protein science.
2. The paper is easy to follow.

# Weaknesses:
1. In Conclusion, while the authors claim that "Results demonstrate the system's capability to generate novel, high-impact hypothese ...", the paper only provides a few examples without presenting a comprehensive evaluation of the system on the entire dataset. Without an overall evaluation metric, it is difficult to assess the approach's reliability and effectiveness.
2. This work employs many agents, but the authors do not specify their sources, thus making it unclear how these agents are obtained, configured, or validated, which raises concerns about reproducibility and reliability.

# Typo:

Section 2.2.1: belwo &rarr; below

---

### Official Review · Reviewer_LDTB · 2025-03-02
**Review of ProteinHypothesis**

**Rating:** 6
**Confidence:** 4

**Review:**

Summary:
This study presents an AI-driven multi-agent framework for automated hypothesis generation in molecular biology. Using RAG and structured data, it synthesizes insights, evaluates feasibility via CoT, and validates hypotheses. Results show its ability to generate novel hypotheses in protein stability, enzyme catalysis, and biomolecular interactions, demonstrating AI’s potential in accelerating discovery for drug development, synthetic biology, and protein engineering.

Strengths:
1) This paper introduces a novel multi-agent framework integrating RAG with structured experimental data.
2) The three-phase framework has a broad applicability. It can generate novel hypotheses across various domains, such as protein stability, enzyme catalysis, and biomolecular interactions.
3) By automating hypothesis generation and validation, the framework has the potential to speed up research processes in biology.

Weaknesses:
1) With RAG, multi-agent LLMs, and additional CoT reasoning, this framework is computationally expensive. This may pose challenges for widespread adoption.
2) The authors did not compare AI-generated hypotheses with those from domain experts, making it unclear how AI performance compares to human-driven scientific discovery.
3) Readability: Consider breaking up some long paragraphs, especially in the introduction section, as they make it difficult for the reader to engage with the content from the start.

---

### Official Review · Reviewer_vVJm · 2025-03-03
**ProteinHypothesis: A Physics-Aware Chain of Multi-Agent RAG LLM for Hypothesis Generation in Protein Science**

**Rating:** 5
**Confidence:** 3

**Review:**

Paper Summary:

The paper hypothesizes that integrating RAG-based literature analysis in a multi-agent evaluation framework can effectively generate scientifically rigorous and experimentally testable hypotheses in protein science. The authors propose that a three-phase approach using specialized AI agents can refine hypotheses through systematic validation, ensuring both scientific merit and practical applicability.

Strengths:
- Domain Specialization: This domain specialization likely improves the relevance and applicability of the generated hypotheses compared
   to more general-purpose hypothesis generation systems.

- Multi-Phase Evaluation Process: The three-phase approach with progressive refinement through different specialized agents provides a
   robust framework for ensuring hypotheses

Weaknesses:
- Unclear Technical Implementation Details: The paper omits crucial information about the LLMs used, their parameters, and fine-tuning approaches, making reproduction difficult and obscuring the computational basis of the system's performance.

- Limited Discussion of Hallucination Risks: While the paper uses RAG to retrieve scientific content, it lacks specific verification mechanisms against hallucinations. For example, how does the RAG system handle the structured information? Despite multi-agent evaluation, the system needs better fact-checking and confidence scoring when extrapolating beyond retrieved information.

- Handling Novel Scientific Domains: How does your system perform in emerging protein science subfields with sparse literature or when studying protein families with limited data? What mechanisms address knowledge gaps in these scenarios?

Questions to Authors:

See the above

---

### Decision · Program_Chairs · 2025-03-05

Accept (Poster)